# Phenotyping of a rice (*Oryza sativa L*.) association panel identifies loci associated with tolerance to low soil fertility on smallholder farm conditions in Madagascar

**Juan Pariasca-Tanaka[1], Mbolatantely Fahazavana Rakotondramanana[2], Sarah Tojo Mangaharisoa[2], Harisoa Nicole Ranaivo[2], Ryokei Tanaka[3], Matthias Wissuwa**[1] *

**1** Crop, Livestock and Environment Division, Japan International Research Center for Agricultural Sciences (JIRCAS), Tsukuba, Japan, **2** Rice Research Department, The National Center for Applied Research on Rural Development (FOFIFA), Antananarivo, Madagascar, **3** Department of Agricultural and Environmental Biology, Graduate School of Agricultural and Life Sciences, The University of Tokyo, Tokyo, Japan

* wissuwa@affrc.go.jp

## Abstract

Rice (*Oryza sativa L*.) is a staple food of Madagascar, where per capita rice consumption is among the highest worldwide. Rice in Madagascar is mainly grown on smallholder farms on soils with low fertility and in the absence of external inputs such as mineral fertilizers. Consequently, rice productivity remains low and the gap between rice production and consumption is widening at the national level. This study evaluates genetic resources imported from the IRRI rice gene bank to identify potential donors and loci associated with low soil fertility tolerance (LFT) that could be utilized in improving rice yield under local cultivation conditions. Accessions were grown on-farm without fertilizer inputs in the central highlands of Madagascar. A Genome-wide association study (GWAS) identified quantitative trait loci (QTL) for total panicle weight per plant, straw weight, total plant biomass, heading date and plant height. We detected loci at locations of known major genes for heading date (hd1) and plant height (sd1), confirming the validity of GWAS procedures. Two QTLs for total panicle weight were detected on chromosomes 5 (*qLFT5*) and 11 (*qLFT11*) and superior panicle weight was conferred by minor alleles. Further phenotyping under P and N deficiency suggested *qLFT11* to be related to preferential resource allocation to root growth under nutrient deficiency. A donor (IRIS 313–11949) carrying both minor advantageous alleles was identified and crossed to a local variety (X265) lacking these alleles to initiate variety development through a combination of marker-assisted selection with selection on-farm in the target environment rather than on-station as typically practiced.

## Introduction

Rice (*Oryza sativa L*.) is a staple food for more than half of the world population, supplying about 35 to 60% of dietary calorie intake, micronutrients (Fe and Zn) and vitamins (B). In

**Data Availability Statement:** All relevant data are within the manuscript and its Supporting information files. The genotype dataset is publicly available at https://snp-seek.irri.org/_download.zul.

**Funding:** This research was funded by the Science and Technology Research Partnership for Sustainable Development (SATREPS), Japan Science and Technology Agency (JST)/Japan International Cooperation Agency (JICA) (Grant No. JPMJSA1608). The funders had no role in study design, data collection and analysis, decision to publish, or preparation of the manuscript.

**Competing interests:** The authors have declared that no competing interests exist.

Madagascar, an island located off the southeast coast of Africa, rice has been introduced during the early migration from Asia [1,2] and remains the dominant staple in the Malagasy diet. The annual per-capita consumption of about 136 kg rice is among the highest in the world but unfortunately local production cannot meet the rising demand of an increasing population [3]. The situation is similar across the Sub-Saharan Africa (SSA) region where rice consumption is rapidly outpacing local production [4].

Rice productivity in most of the SSA region is limited by several biotic and abiotic stresses. Among the abiotic stresses, low soil fertility is the one of main concern, with phosphate (P) often being the most limiting nutrient [5]. This is certainly the case in Madagascar where P deficiency is widespread, possibly due to high levels of P-fixing element such as iron (Fe), aluminum (Al) and/or oxyhydroxides [6]. This problem is exacerbated by the continual removal of organic residues and limited application of organic matter and/or fertilizer inputs [7–9]. The nutritional deficiency in soil could be alleviated through fertilizer application; however, the cost of fertilizers is often higher in SSA compared to other regions and therefore access to fertilizers is limited for resource-poor farmers in small scale farming systems. Approximately 10% of the global population lives in Africa, however, only 0.8% (1.29 TM) of the total amount of applied fertilizer is used in Africa [10].

A cost-efficient partial solution to the soil fertility problem in SSA would be the improvement of nutrient acquisition and utilization efficiencies in local varieties [11]. Studies evaluating gene bank accessions concluded that ample variation for P acquisition and underlying root traits existed in the rice gene pool, with traditional varieties typically being superior to modern high-yielding varieties [12]. In comparison, less variation was observed within rice accessions for internal P utilization efficiency, but again traditional varieties tended to be more efficient [13]. The prevalence of traditional rice varieties throughout Madagascar [14] would confirm that modern varieties lack adaptation to low-input conditions and, furthermore, may suggest that plant breeding has not properly addressed the needs of the mostly resource-poor smallholder farmers.

Gene banks are considered a reservoir of untapped allelic variants waiting to be utilized for improving crop adaptation to biotic and abiotic stresses [15,16]. Through next generation sequencing (NGS) an increasing number of gene bank accession has been sequenced, leading to the detection of allelic variants mainly as single nucleotide polymorphisms (SNPs) where the genome sequence of two or more individuals differs by a single base. Genome-wide association studies (GWAS) detect associations between a genetic variant (SNPs throughout the genome) and trait variation (phenotype) for a large number of individuals. GWAS could identify loci, genes and alleles that contribute to specific traits, and therefore could accelerate breeding for targeted traits. For example, GWAS had been successfully applied in rice to dissect the genetic basis of nutrient-related traits such as aluminum tolerance [17], phosphorus utilization efficiency [12], manganese toxicity [18], sulfur deficiency [19], and of traits related to root development [20], and root efficiency [12].

Accessions combined in the GWAS panels of above studies typically comprise mostly traditional varieties from gene bank collections but may also include modern varieties and breeding lines. They may focus on accessions of just one rice sub-species or include representatives of all or most sub-species. Thus, GWAS represents a structured approach to assess the genetic diversity held at national/international gene banks (that would otherwise not be utilized), potentially identifying novel donors and alleles to be utilized in crop breeding to improve traits that lack genetic variability in the pool of currently used breeding lines.

The objective of this study is to follow such an approach with the goal to improve adaptation of rice to low soil fertility. A GWAS panel of 532 sequenced rice accessions was imported from IRRI and evaluated on-farm under low-input conditions in the central highlands of

Madagascar in order to identify novel loci associated with traits of relevance in such conditions. For loci detected we aim to identify suitable donors to initiate a local breeding program to improve grain yield for smallholder farmers.

## Materials and methods

### 1. On-farm trials

**1.1 Plant material and on-farm experiment.** The 3000 Rice Genome Project (3KRGP) housed by the International Rice Research Institute, IRRI-Philippines (http://snp-seek.irri.org) [21], provides publicly available genotype information and seeds of sequenced accessions. A set of 532 rice accessions from this resource was imported from IRRI to Madagascar. The selected set predominantly included accessions from the *indica* subpopulation (81%), with minor proportions from the *japonica*, *aus* and *aromatic* subspecies (Fig 1A). The central focus in *indica* group was because of the preference for *indica*-type varieties by farmers and consumers in Madagascar. Accessions were then selected from different rice producing countries with similar conditions to the Central Highlands of Madagascar such such as India, Lao PDR, Thailand, Indonesia, Nepal, Sri Lanka, Bangladesh, Philippines, (Fig 1B). Several accessions originated from Madagascar were also included in the set.

Trials were conducted at two field sites in the highlands of Madagascar during the main rice growing season (November to April, 2017–2018): Anjiro-Moramanga (Latitude: -18˚ 56' 58.13" S, Longitude: 48˚ 13' 48.25"), at 950 m above sea level (masl), with average maximum temperature of 29˚C and average minimum temperature of 17˚C, and average precipitation of 220 mm; and Ankazo-Antsirabe (Latitude: -19˚ 51' 57.10" S, Longitude: 47˚ 01' 59.99", at 1150 masl, with average maximum temperature of 27˚C and average minimum temperature of 15˚C, and average precipitation of 210 mm.

All experiments were conducted on smallholder farms characterized by low-input cultivation. Field plots used had no history of mineral fertilizer application in the past. The soils used in these experiments were clay loam, with pH: 5.3–5.8 (1:5, $H_2O$), total N (g kg$^{-1}$): 0.4–0.6 g kg-1, Olsen P (mg kg$^{-1}$): 5.9–7.5, and organic C (g kg$^{-1}$): 10.7–15.3. Seeds were sown in elevated nursery beds (20 m L x 0.6 m W x 0.1 m H). Each accession was sown in a 40 cm row with 10 cm spacing between rows (S1 Fig). Seeds were covered with fine soil and a layer of non-rice straw mulch to maintain moisture and warmness during seed germination. Water was supplied (depending on availability) either by partially flooding the bed soil or manually using watering cans. Seedlings were raised for 4 weeks in this nursery, followed by manual transplanting of 1 plant per hill into 2-m long single-row plots with 20 cm spacing within and between rows. The experiment was conducted in a completely randomized block design (CBRD) with two replications per site. Agronomic practices such as manual weeding, watering, etc, were performed following the local practices.

**1.2 Phenotypic data.** Phenotypic parameters evaluated in this experiment included heading date, plant height and culm height, straw and panicle weight. Heading date were taken throughout the vegetative period (start, 50% and 100%). Plant height was defined as the distance from the base of the stem to the tip of the flag leaf, while culm height was measured up to the panicle base node. Harvests of panicles and straw were done continuously as plants reached maturity, using 5 plants per plot. Panicles were individually harvested by cutting at the basal node of the rachis. Straw weight was recorded as fresh weight directly at the field site. This was later adjusted to dry weight based on the moisture content determined for a sub-sample after drying samples in an oven for 3 days at 60˚C. Harvested panicles were brought to a green house and air dried in mesh bags before weights were taken.

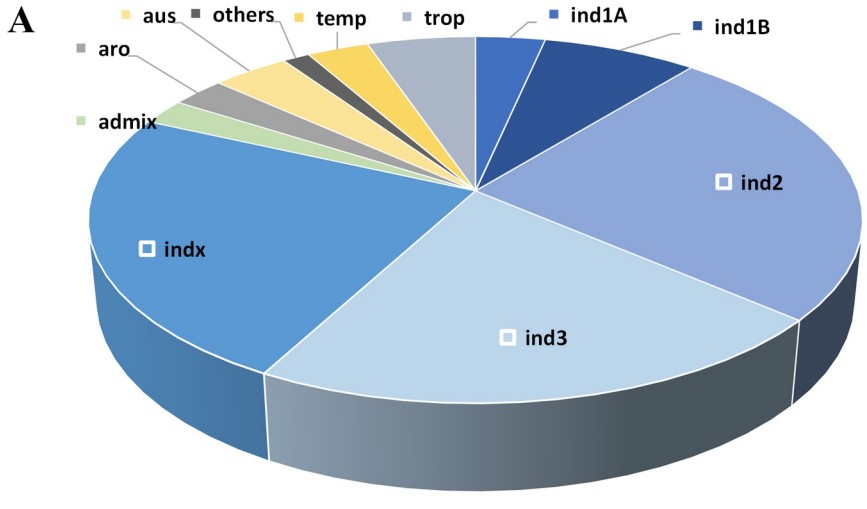

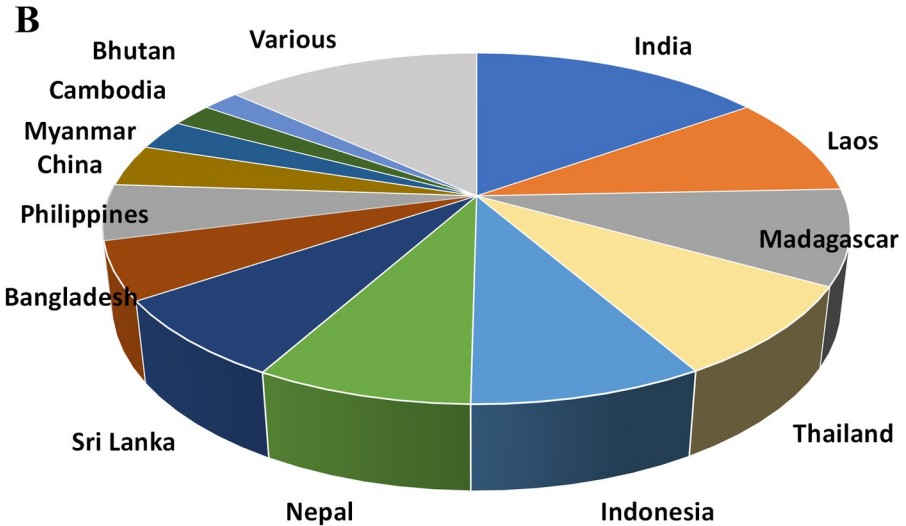

**Fig 1. Distribution of subspecies (A) and country of origin (B) of selected rice accessions in the GWAS panel.** aro: aromatic; aus: aus; ind: indica; trop: tropical; temp: temperate; admix: admixture.

The phenotypic data was analyzed using a mixed linear model where the effects of genotype were considered as fixed effect, and those of locations and replicates per location as random effects. The analysis was performed using the R package Linear Mixed-Effects Models using 'Eigen' and S4 (lme4) [22], and an in-house script based on:

$$[\text{phenotypic value}]_{\text{ilrj}} = [\text{genotypic value}]_{\text{i}} + [\text{location}]_{\text{l}} + [\text{replication}]_{\text{lr}} + [\text{error}]_{\text{ilrj}}$$

This analysis provided the best linear unbiased estimator (BLUE) for genotypes across 2 sites and 2 replicates per site.

Heritability was estimated by using the following model:

$$y_{ijkl} = \mu + genotype_i + location_k + rep(\text{year} \times location)_{jkl} + e_{ijkl}$$

where $y_{ijkl}$ is the phenotypic value of the $i$-th genotype in $j$-th year on the $l$-th replicate in the $k$-th location.

**1.3 Association mapping.** The 404K core SNP genotype dataset of the rice accessions was obtained from the 3000 Rice Genome Project (3KRGP), (https://snp-seek.irri.org). A matrix genotype file composed of 186,229 (187K) SNPs and 3026 accessions was prepared and reported in a previous study [23]. A subset containing the 532 accessions was filtered from the matrix prior to analysis. Association analysis was then performed using: a) BLUE values obtained for each trait; b) the 187K matrix genotype dataset; and, c) the GWAS function in the Ridge Regression and Other Kernels for Genomic Selection package (rrBLUP v.4.6) [24], using a simple mixed model, where in the phenotype was estimated by setting the accession and residual effects as random, while the replicate effect considered as fixed effect. The effect of population structure was controlled by using a genomic relationship matrix calculated in the A.mat function. The model was run using an in-house R script and the rrBLUP package returned a quantile-quantile plot and a Manhattan plot with a significant threshold set to a 5% FDR (false discovery rate) [20,23]. Loci were considered significantly associated with a trait based on a threshold of $-\log(10)(p) > 5$ for those peaks characterized by at least three consecutives Quantitative Trait Nucleotide (QTN) [19,20].

For the purpose of confirming detected associations, a second analysis was conducted with the software program Trait Analysis by association, Evolution and Linkage 5.0 (TASSEL) [25]. Prior to association analysis, the 404K coreset SNP genotype dataset was filtered as follows: heterozygotes and indels were set as missing values, and SNP having more than 5% missing data or minor allele frequency (MAF) below 0.03 were excluded [19]. The association mapping was then analyzed using the mixed linear model (MLM) procedure, with three principal components (PCA) and a kinship matrix. Adjusted p-values were calculated using the False Discovery Rate (FDR = 0.05) correction method in R.

The phenotypic effect of minor alleles at each locus was determined by calculating the average phenotypic values of all accessions carrying either allele, and a box-plot graph was generated for each locus using an in-house R script.

Linkage disequilibrium (LD) analysis to define LD blocks (non-random association of alleles at a defined region) surrounding the significant SNPs was performed by Haploview 4.2 [26]. To check whether identified regions corresponded to previously identified QTL, peak QTN positions were searched against the public QTL databases QTARO (http://qtaro.abr.affrc.go.jp) and GRAMENE (https://archive.gramene.org/qtl).

**1.4 Selection of putative candidate genes.** Gene models were obtained from the Rice Annotation Project Database (RAP-DB, https://rapdb.dna.affrc.go.jp/) for each significant peak and their surrounding LD block. Genes annotated as '(retro)transposon', 'hypothetical' or 'unknown' were excluded from further analysis. Putative candidate genes were then selected based on annotated function and gene ontology (http://www.geneontology.org), and expression pattern obtained from the Rice XPro database (http://ricexpro.dna.affrc.go.jp).

Furthermore, SNP variant effects were investigated using the Variant Effect Predictor (VEP, Ensembl, https://asia.ensembl.org/Tools/VEP), which predicts the potential effects of the SNP variant in terms of changes in protein sequences.

## 2. Validation of result

**2.1 Using a different set of 3K accessions.** To confirm the effects of positive alleles identified in the on-farm trials (experiment 1), a different set of 3K accessions was grown in the following year and TPW was measured. This set consisted of 52 newly imported rice accessions and 23 accessions repeated from year 1. Field experiments were carried out at the same sites as

in year 1 but in different small-holder farmer fields (under low fertility soil). Fields were not fertilized and had no history of mineral fertilizer application in the past. Experimental procedures were as reported for experiment 1.

**2.2 Using water culture (low P and/or low N).** A second confirmatory experiment was conducted in the greenhouse in Japan, evaluating accessions at the vegetative growth stage in hydroponic culture. Dehulled seeds from selected accessions had been imported into Japan where a seed multiplication step was necessary. Pregerminated seeds were sown onto a mesh floating over a solution containing 10% Yoshida solution without P. The full-strength Yoshida solution (1X) is composed of: N, 2.86 mM (as $NH_4NO_3$); P, 0.05 mM; K, 1mM; Ca, 1mM; Mg, 1mM; Mn, 9 μM; Mo, 0.5 μM; B, 18.5 μM; Cu, 0.16 μM; Fe, 36 μM; Zn, 0.15 μM [27].

Ten days after germination seedlings were transferred to 45-L hydroponic containers with 28 seedlings fixed to holes in the container lid using sponge strips. Four treatments were imposed in an otherwise modified Yoshida nutrient solution as described above: low P (LP, 5uM), low N (LN, 0.28 mM), a combination of low N and low P (LNP) and a control treatment (2.86 mM N, 50uM P). The experiment was conducted in a temperature-controlled (30˚C during daytime, and 25˚C nighttime) glass house under natural light at JIRCAS-Tsukuba (36˚12'0"N, 140˚6'0"E). The experiment was conducted in a randomized complete block design (RCDB) with four replications. Rice accessions were harvested 35 days after germination, root length was evaluated together with root and shoot dry matter, and root/shoot subsamples of four independent replications were flash-frozen in liquid nitrogen and stored at -70˚C until RNA extraction using the RNeasy Plant Mini Kit (Qiagen), following the manufacturer instruction manual.

*Expression of candidate genes*. Total RNA (400 ng) was then reverse transcribed (RT) using the PrimeScript RT Enzyme Mix I (Takara, Japan). Quantitative PCR (qPCR) was performed using 2 ng RT template and SYBR Premix ExTaq (Perfect Real Time, Takara, Japan), using the CFX96 Touch Real-Time PCR system (BioRad, USA). Primer efficiency was determined by serial dilutions of RT product. Elongation factor (ELF-1), Glyceraldehyde 3-phosphate dehydrogenase (GAPDH) and Ubiquitin (Ubi) was used as internal controls. Relative expression levels between treatment and shoot or root of control samples were calculated using the standard-curve method and expressed as fold changes. The normalized data was then analyzed by ANOVA. The list of primers used in this study is shown in S1 Table.

## 3. Development of a cross population

In order to utilize the main peaks associated with TPW on chromosome 11 in marker assisted selection, we designed a Kompetitive Allele Specific PCR (KASP) marker (qTLF11-1). Using this KASP marker we determined that the popular local Malagasy variety X265, also known as "Mailaka", carries the (major) unfavorable allele, and would therefore be a potential recipient benefitting from the introgression of the positive minor allele from donor accession GP1103.

X265 and GP1103 parent plants were grown under paddy condition and during flowering time, panicles of previously designated female plants were emasculated using heat treatment (immersion in water bath at 42˚C for 7 min) and cross-pollinated using pollen from the male parent. Successfully crossed F1 plants were identified with KASP markers using an in-house protocol [28] following the manufacturer instruction manual (LGC Genomics). In brief, KASP amplification was performed using allele-specific primers with FAM and HEX fluorophores, a common primer and master mix. The fluorescence signal was then recorded at 520 nm (FAM) and 556 nm (HEX) for 2 min at 25˚C, at the end of the thermal cycles.

Using a modified rapid generation advance (RGA) protocol, crossed individuals were advanced through the F2 and F3 generation and F4 seeds were sent to Madagascar for field evaluations in on-farm trials.

**Statistical analysis.** The effects of treatment, allele and their interaction on different traits were estimated using a one or two-way ANOVA, and mean comparisons were performed using Tukey's honestly significant difference (HSD) post hoc test (Statistix 9.0 Software). Correlation values were generated by "Hmisc" and visualized in scatter plots by "PerformanceAnalytics" R packages [29].

## Results

### On-farm field trials and GWAS analysis

A set of rice accessions of diverse origin but primarily belonging to the *indica* sub-species (Fig 1) was imported and evaluated in two rice-growing areas in the central highlands of Madagascar. Plant performance was evaluated by straw dry weight (STW), total panicle weight (TPW), which is the average total weight of all panicles per one plant (rather than the weight of an individual panicle) and total dry weight (TDW) (Fig 2). Biomass weights are given per plant and grain yield is estimated by TPW. STW ranged from 7.1 to 97.4 g plant$^{-1}$ with a mean of 29.5 g plant$^{-1}$, TPW from 3.0 to 41.5 g plant$^{-1}$ with a mean of 16.3 g plant$^{-1}$, and TDW from 10.1 to 121.1 g plant$^{-1}$ with a mean of 45.2 g plant$^{-1}$ (Fig 2, S2 Table). Traits STW and TPW showed greatest variation with a coefficient of variation (CV) of 38.8 and 33.7%, respectively, which indicates great accession variability in their adaptation to the new environment. The distribution for TPW was near-normal with the exception of three outliers with high values.

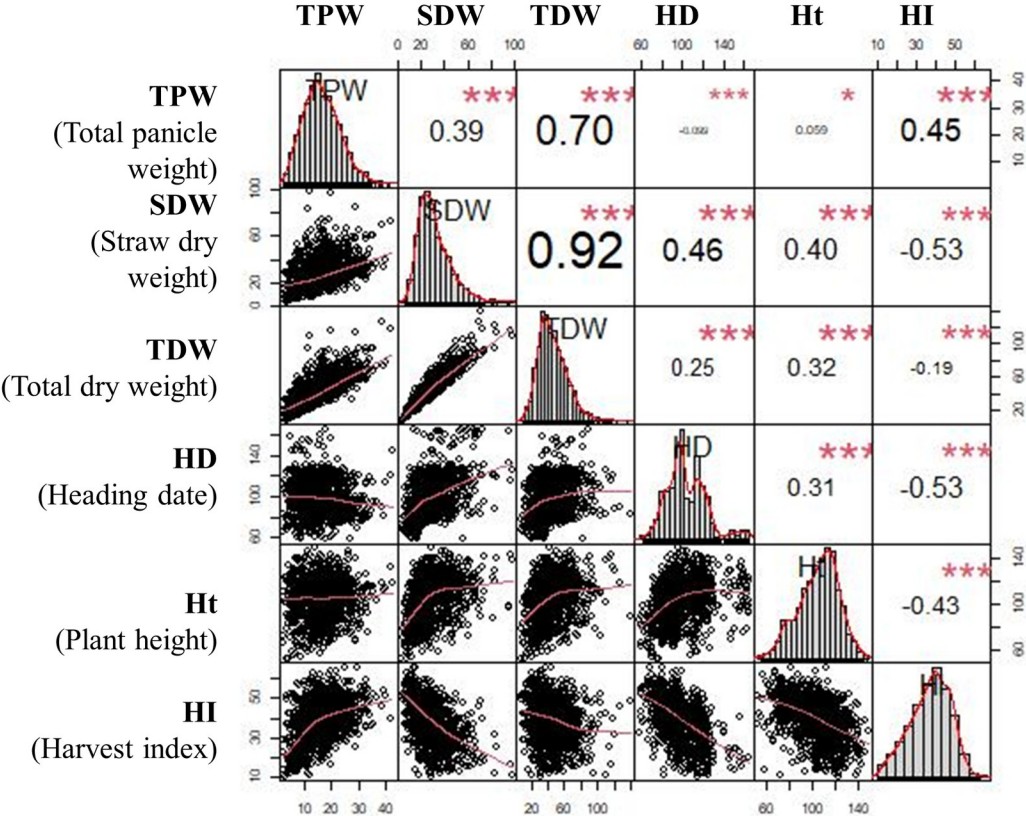

**Fig 2. Scatterplots showing the relationship among all evaluated agronomic traits.** The distribution of each variable is shown on the histogram, while the bivariate scatter plots with a fitted line, and correlation values and significance level are shown on the left and right side, respectively. Significance level: p-values (p<0.001: ***, p<0.01: **, p<0.05: *, p>0.05: "").

The distribution for STW was slightly skewed towards smaller values with five accessions showing high STW.

Correlation coefficients between traits measured ranged from as high as r = 0.92 between TDW and STW, to as low as 0.06 between TPW and plant height (Fig 2). Straw and panicle biomass had a moderately positive correlation and variation in STW contributed more to TDW compared to TPW (r = 0.70). As expected, STW was positively correlated with plant height (r = 0.40) but plant height did not affect TPW. Similarly, late heading was associated with higher STW (r = 0.46) but not with TPW. Heading date (HD) was negatively correlated with harvest index (HI), presumably because late heading accessions produced more straw biomass (Fig 2).

Higher heritability values were found for SWT and HD with 0.54 and 0.40, respectively, while TPW showed a value of 0.30 (S2 Table).

The association analysis using the Mixed Linear Model (MLM) in rrBLUP identified several quantitative trait loci (QTL) associated with tolerance of low-fertility soils (qLFT) (Fig 3). Two QTLs associated with TPW were detected on chromosomes 5 (qLFT-5) and 11 (qLFT-11). These loci were represented by two significant Quantitative Trait Nucleotide (QTNs) at 14.496 and 14.827 Mbp on chromosome 5 and by 3 QTNs between 25.827–25.849 Mbp on chromosome 11 (S3 Table). For both loci the minor allele frequency (MAF) was below 10% and the minor allele had a positive effect, increasing TPW from 15.9 to 22.7 g plant$^{-1}$ (+42.8%, chromosome 5) and from 15.8 to 22.0 g plant$^{-1}$ (+39.0%, chromosome 11) (Tables 1 and 2).

Four QTLs were detected for STW on chromosomes 1, 3, 4 and 11 (Fig 3). The strongest effect was seen on chromosome 1 where four consecutive QTNs between 10.993 and 11.591 Mbp exceeded the significance threshold of 5 (S3 Table). This locus had a MAF of 5% and the minor allele was estimated to increase STW from 28.5 to 46.5 g plant$^{-1}$ (+63.2%) (Tables 1 and 2). The second most significant QTL for STW was delineated by 3 QTN between 31,542 and 31,543 Mbp on chromosome 4 and the minor allele (MAF = 12%) increased STW by 42.7% (Table 1). The remaining two QTL on chromosomes 3 and 11 had lower significance but estimated phenotypic effects were large with 61% and 54% increase in STW due to the minor allele, respectively (Table 1). Two QTLs associated with TDW were detected on chromosome 5 and 11 but at different locations from QTL for TPW and STW and with lower significance (Fig 3). Unlike for above loci, the MAF was not low but above 30%. For the locus on chromosome 11 the minor allele increased TDW by 20% but a negative effect was associated with the minor allele on chromosome 5 (Table 1).

Additionally, QTLs for heading date (HD) were found in chromosome 1, 2, 3, 4, 6 and 7 (Table 1 and S3 Table) with the most significant association detected on chromosome 6 at 21,342 Mbp where the minor allele delayed heading by 30%. For plant height a highly significant locus was detected between 37,876–39,548 Mbp on chromosome 1. This interval contains the known semi-dwarf gene *sd1* and the minor allele reduced plant height by 30%.

The result was then validated using the software TASSEL with the 3K-400K SNP dataset. The resulting Manhattan and QQ plots for each evaluated trait are shown in Table 1, and S2 Fig.

The MAF values calculated for the studied panel were corroborated for the entire 3K dataset. For the two QTL associated with TPW (5@14,496,649 and 11@25827214) the frequency of minor alleles was below 10% in the entire 3K set (S4 Table) and therefore very similar to the subset phenotyped in Madagascar (Table 1). More than 90% of the accessions with the minor allele belong to the *indica* sub-species (ind1, ind2, ind3, and indx). For loci associated with STW (1@ 11039294, 3@ 20761847, and 4@31542322) a very similar situation was observed, the minor allele predominantly being detected in the indica group.

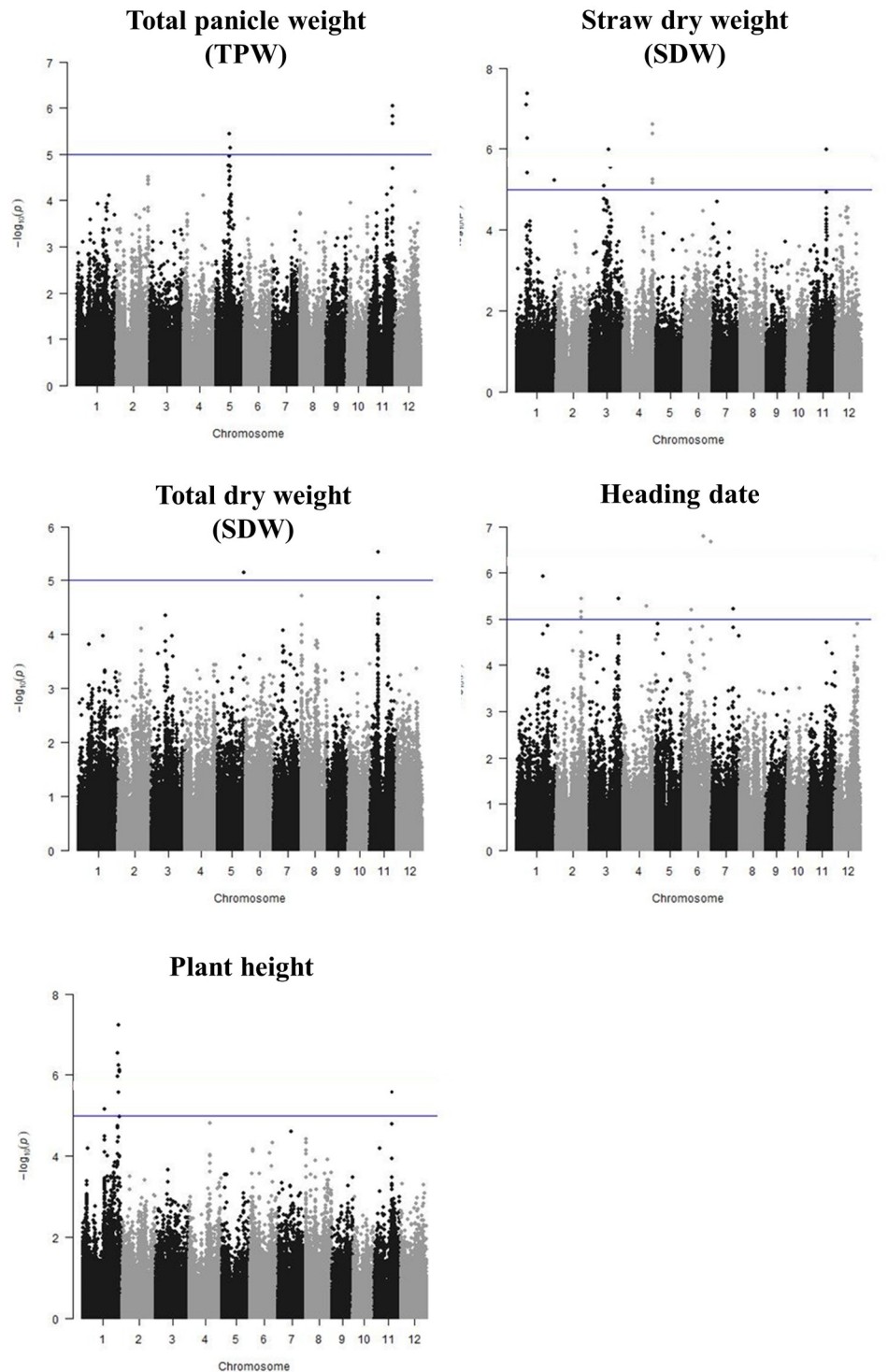

**Fig 3. Manhattan plots derived from GWAS analysis for all evaluated traits.** Y-axis shows the negative logarithm of the association ((-log10 (P-value)) for each SNP, while X-axis displays the SNP location along the 12 chromosomes. Vertical line indicates a -log10 (P-value) threshold of 5.

**Table 1. Summary of quantitative trait loci (QTL) associated with low fertility soil for several agronomic traits using a mixed linear model (MLM).**

| Trait | Loci name | Chr | SNP denomination | SNP position | P value | | minor allele | |
|---|---|---|---|---|---|---|---|---|
| | | | | (bp) | rrBLUP | [3]TASSEL | [1]MAF | [2]effect |
| **Total panicle weight (TPW)** | | | | | | | | |
| | qLFT-5 | 5 | 5@14496649 | 14,496,649 | 3.5E-06 | 3.48E-05 | 0.04 | 41 |
| | qLFT-11 | 11 | 11@25827214 | 25,827,214 | 2.1E-06 | 4.23E-05 | 0.08 | 39 |
| **Straw dry weight (SDW)** | | | | | | | | |
| | qLFT-1 | 1 | 1@11039294 | 11,039,294 | 4.2E-08 | 4.32E-05 | 0.05 | 60 |
| | qLFT-3 | 3 | 3@20761847 | 20,761,847 | 1.0E-06 | 5.03E-05 | 0.06 | 60 |
| | qLFT-4 | 4 | 4@31542322 | 31,542,322 | 4.3E-07 | 1.44E-05 | 0.11 | 40 |
| | qLFT-11s | 11 | 11@19334313 | 19,334,313 | 1.0E-06 | 3.95E-05 | 0.04 | 54 |
| **Total dry weight (TDW)** | | | | | | | | |
| | LFT-11t | 11 | 11@8850567 | 8,850,567 | 3.0E-06 | 8.2E-05 | 0.34 | 20 |
| **Heading date (HD)** | | | | | | | | |
| | qHD-1 | 1 | 1@28488608 | 28,488,608 | 1.2E-06 | 2.2E-05 | 0.10 | 30 |
| | qHD-2 | 2 | 2@27125471 | 27,125,471 | 3.5E-06 | 6.7E-05 | 0.08 | 23 |
| | qHD-3 | 3 | 3@31256576 | 31,256,576 | 3.6E-06 | 5.1E-05 | 0.30 | 21 |
| | qHD-4 | 4 | 4@25834920 | 25,834,920 | 5.4E-06 | 4.9E-05 | 0.05 | 30 |
| | qHD-6 | 6 | 6@21342504 | 21,342,504 | 1.6E-07 | 3.4E-05 | 0.07 | 30 |
| | qHD-7 | 7 | 7@22523621 | 22,523,621 | 6.2E-06 | 7.1E-05 | 0.32 | 21 |
| **Plant height (Ht)** | | | | | | | | |
| | qHt-1 | 1 | 1@38730952 | 38,730,952 | 6.0E-08 | 1.6E-05 | 0.11 | -30 |

[1] MAF: minor allele frequency.

[2] allele effect: phenotypic value (((minor allele-major allele)/major allele)*100).

[3] values corrected by False discovery rate (FDR).

For main QTL associated with TPW and STW, minor and major alleles were investigated in detail in relation to effects on other traits (Table 2). Accessions belonging to the minor allele group for TPW QTL qLFT-5 and qLFT-11 had similar HD and STW compared to the group with the major allele. Accessions carrying minor alleles at both loci (n = 9) showed a further improvement in TPW, being 82.1% superior to accessions lacking both loci. For STW QTL, the minor and major allele groups did not differ significantly for TPW, but the minor allele group showed significantly later heading (Table 2). We also calculated effects of having two of these loci simultaneously and while this led to further increases in STW, it caused additional delays in heading and several accessions were not yet mature at the end of the experimental period (data not shown).

**Selection of putative candidate genes.** To determine to what distance linkage would extend from the peak QTN, the relatedness of all SNP in the larger region surrounding the peak QTN were investigated (S3 Fig). Very distinct linkage blocks could not be identified but based on the decay in LOD between markers we identified likely regions to be considered for candidate gene identification for qLFT-5 from 14.343 to 14.585 Mbp, and from 25.734 to 25.948 Mbp for qLFT-11. Potential candidate genes for TPW were selected based on their expression pattern in different tissues and environmental conditions (RiceXpro, S4 Fig) and their functional annotation is listed in S5 Table (excluding unknown genes and hypothetical proteins). Estimating functional consequences of SNPs in candidate genes using the Variant Effect Predictor (Ensembl) showed that most SNPs for TPW were located in the intergenic,

**Table 2. Distribution and interaction of minor and major alleles across the main identified QTLs.**

| | | minor allele | | major allele | | effect | |
|---|---|---|---|---|---|---|---|
| | | mean | SD | mean | SD | % | |
| **Total panicle weight (TPW)** | | | | | | | |
| *qLFT-5* | TPW | 22.7 | 8.3 | 15.9 | 5.1 | 42.8 | *** |
| | HD | 95.3 | 15.3 | 100.1 | 16.6 | -4.8 | ns |
| | STW | 31.4 | 10.7 | 28.4 | 10.4 | 10.6 | ns |
| | n | 27 | | 461 | | | |
| *qLFT-11* | TPW | 22.0 | 7.7 | 15.8 | 5.2 | 39.2 | *** |
| | HD | 97.4 | 18.6 | 100.0 | 16.5 | -2.6 | ns |
| | STW | 29.3 | 10.8 | 28.5 | 10.4 | 2.8 | ns |
| | n | 32 | | 447 | | | |
| *qLFT- 5 x 11* | TPW | 28.4 | 9.1 | 15.6 | 5.1 | 82.1 | *** |
| | HD | 94.1 | 22.2 | 100.3 | 16.7 | -6.2 | ns |
| | STW | 30.0 | 13.3 | 28.3 | 10.5 | 6.0 | ns |
| | n | 9 | | 430 | | | |
| **Straw dry weight (STW)** | | | | | | | |
| *qLFT-1* | STW | 46.5 | 17.9 | 28.5 | 10.2 | 63.2 | *** |
| | HD | 125.7 | 22.0 | 100.3 | 17.5 | 25.3 | *** |
| | TPW | 14.8 | 4.4 | 16.3 | 5.58 | -9.2 | ns |
| | n | 25 | | 486 | | | |
| *qLFT-3* | STW | 45.4 | 16.6 | 28.2 | 10.1 | 61.0 | *** |
| | HD | 124.5 | 22.4 | 99.9 | 17.3 | 24.6 | *** |
| | TPW | 12.9 | 4.3 | 16.5 | 5.6 | -21.8 | ns |
| | n | 30 | | 466 | | | |
| *qLFT-4* | STW | 40.1 | 15.5 | 28.1 | 10 | 42.7 | *** |
| | HD | 115.4 | 20.8 | 99.8 | 17.7 | 15.6 | *** |
| | TPW | 14.6 | 5.6 | 16.5 | 5.5 | -11.5 | ns |
| | n | 59 | | 448 | | | |

Significance levels (***, **, *, ns: $p < 0.001$, 0.01, 0.05, non-significant, respectively).

SD: standard deviation.

TPW (total panicle weight), STW (straw total weight), HD (heading date).

and up/down stream region (more than 80%), while few existed in the intron or 5/3'UTR regions (S5 Fig). Two genes had either gained or lost a stop codon but none of these were considered functionally relevant.

Based on above criteria the following potential candidate genes for panicle weight at *qLFT-5* and *qLFT-11* were identified: 1-aminocyclopropane-1-carboxylic acid synthase (Os05g0319200), protein kinase (Os05g0319700), WRKY transcription factor (Os05g0322900), cytochrome P450 (Os05g0320700), and Zn finger protein (Os05g0316000), and NB-ARC domain (Os11g0645886), oxidoreductase (Os11g0645200), E3 ubiquitin-protein ligase EL5 (Os11g0649801), sugar transporter (Os11g0643800) (Table 3 and S5 Table). Candidates for STW would be galactose oxidase (Os01g0300900), and Chitinase precursor (Os01g0303100), while SAM dependent carboxyl methyltransferase family protein (Os11g0260100), polygalacturonase (Os05g0578600), and UDP-glucosyltransferase (Os03g0757000, Os06g0271000) were considered candidate genes for total weight.

**Table 3. List of potential candidate genes in QTLs associated to total panicle weight (TPW), shoot dry weight (SDW) and total dry weight (TDW).**

| RAPdb | MSU (LOC) | Chr | PosMb | Annotation |
|---|---|---|---|---|
| **Total panicle weight, TPW (*qLFS-5*, *qLSF-11*)** | | | | |
| Os05g0316000 | Os05g25180 | 5 | 14.588 | Zinc finger RING/FYVE/PHD-type domain |
| Os05g0319200 | Os05g25490 | 5 | 14.825 | 1-aminocyclopropane-1-carboxylic acid synthase |
| Os05g0319700 | Os05g25540 | 5 | 14.844 | Protein kinase-like protein |
| Os05g0320700 | Os05g25640 | 5 | 14.900 | Similar to Cytochrome P450 |
| Os11g0644800 | Os11g42510 | 11 | 25.597 | Tyrosine/nicotianamine aminotransferases family |
| Os11g0645200 | Os11g42540 | 11 | 25.615 | Oxidoreductase |
| Os11g0645886 | Os11g42590 | 11 | 25.635 | NB-ARC domain containing protein |
| Os11g0648400 | Os11g42850 | 11 | 25.806 | Protein of unknown function DUF3615 domain |
| Os11g0649801 | None | 11 | 25.911 | Similar to E3 ubiquitin-protein ligase EL5 |
| **Straw dry weight (SDW)** | | | | |
| Os01g0300900 | Os01g19480 | 1 | 11.059 | Galactose oxidase |
| Os01g0301000 | Os01g19490 | 1 | 11.065 | Pentatricopeptide repeat domain |
| Os01g0301900 | Os01g19610 | 1 | 11.110 | Protein of unknown function DUF247 |
| Os01g0302500 | Os01g19694 | 1 | 11.167 | Knotted1-type homeobox protein OSH6 |
| Os01g0303100 | Os01g19750 | 1 | 11.209 | Chitinase precurso |
| Os01g0303600 | Os01g19800 | 1 | 11.232 | RING/FYVE/PHD-type domain |
| Os04g0618700 | Os04g52780 | 4 | 31.421 | Protein kinase |
| Os04g0619400 | Os04g52840 | 4 | 31.463 | Protein kinase |
| Os04g0620400 | Os04g52940 | 4 | 31.532 | SIT4 phosphatase-associated protein |
| **Total dry weight (TDW)** | | | | |
| Os05g0578600 | Os05g50260 | 5 | 28.802 | Similar to Polygalacturonase PG2 |
| Os05g0578900 | Os05g50270 | 5 | 28.818 | GAGA-type zinc finger transcription factor |
| Os11g0260100 | Os11g15340 | 11 | 8.677 | SAM dependent carboxyl methyltransferase |
| Os11g0260200 | Os11g15370 | 11 | 8.702 | Sulfotransferase domain containing protein |

## Validation of the TPW QTL

A set of 75 accessions including 52 not previously phenotyped accessions was tested under similar condition as for Experiment 1. Of these 52 new accessions, 21 harbored the positive minor allele at 11_25827214 (*qLFT-11*). This group had significantly higher total panicle weight compared to the group with the major but disadvantageous allele (Table 4). Although both groups showed similar mean values for plant height and number of panicles.

A subset of rice accessions with contrasting alleles at *qLFT-11* was grown in hydroponics under low N (LN), low P (LP) or combined low N and P (LNP) conditions to simulate the low fertility of soils in Madagascar. All nutrient deficient treatments increased root biomass and

**Table 4. Total panicle weight from rice accessions selected from within and outside the GWAS panel.**

| | Number of accessions (n) | Plant height (Ht) | | Number of Panicles | | Total panicle weight (TPW) | |
|---|---|---|---|---|---|---|---|
| Allele (A) | | ns | | ns | | * | |
| Advantageous | 23 | 84.61 | a | 9.69 | a | 35.65 | a |
| Disadvantageous | 52 | 82.22 | a | 9.61 | a | 32.49 | b |

Plants were grown on-farm field, in the next cropping season, under low input condition, Madagascar (Experiment 2–1). The accessions were divided into two groups: harboring the advantageous or disadvantageous alleles for total panicle weight (TPW). Values are the mean of four independent replication. Statistical significance was determined by one-way ANOVA and Tukey's tests. Significance levels (***, **, *, ns: p<0.001, 0.01, 0.05, non-significant, respectively).

this effect was more pronounced in the group harboring the positive minor allele at 11@25827214 (Fig 4, S6 Table). Both groups did not differ significantly for root biomass in the nutrient-replete control treatment but root biomass more than doubled for the minor allele group whereas it increased between 53–59% in the group with the major allele. Shoot biomass, on the other hand, decreased in all nutrient deficient treatments relative to the control (Fig 4). Differences between allelic groups were small and not specific to nutrient deficiency. However, significant differences between groups were seen in the root to shoot ratio, which increased

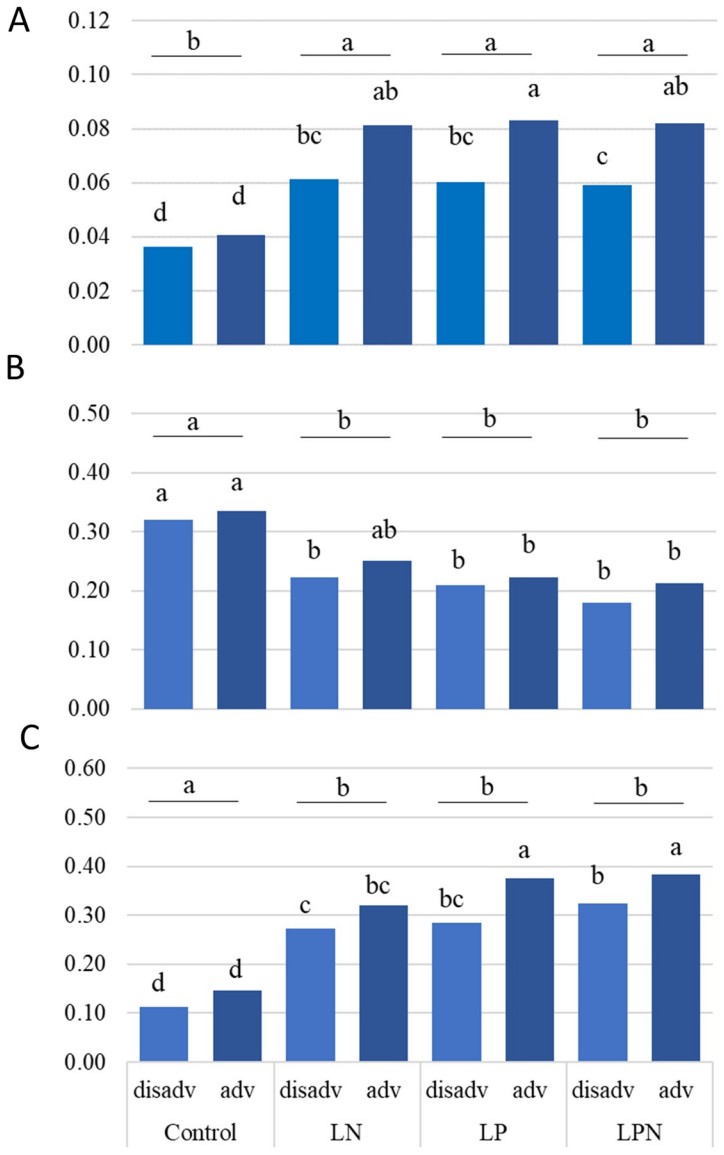

**Fig 4. Root (A), shoot (B) and root to shoot ratio (C) from rice accessions harboring the advantageous or disadvantageous alleles for total panicle weight (TPW).** Plants were grown under hydroponic condition with low Nitrogen and/or low Phosphorus (Experiment 2–2). Values are the mean of four independent biological replicates (n = 4). Statistical significance was determined using two-way ANOVA and Tukey's tests. Different letters represent distinct means within groups at p < 0.05 (***, **, *, and ns refers to p<0.001, 0.01, 0.05, non-significant, respectively). adv: advantageous allele (G-38, G-355, G-1103), disadv: disadvantageous allele (X265, IR64, G-61, G-97).

significantly in all nutrient deficient treatments and for which allelic differences were significant in the two low-P treatments but not in the LN treatment.

Gene expression in shoot and root tissue of the allelic groups under LP, LN and LNP compared to control conditions (base = 1) are represented in a heatmap graph (Fig 5). A gene known to respond strongly to P deficiency (OsSPX) was included to corroborate and gauge the typical P response. This gene showed the highest transcript abundance in low P tissue with no difference between the allele group (Fig 5).

Candidate genes for *qLFT-11* exhibited differential expression across treatments. The genes encoding for oxidoreductase (Os11g0645200), and plant resistance (Os11g0645400) were more responsive to N than to P deficiency and expression tended to be higher in the advantageous allele group. The sugar transporter (Os11g0643800) was only differentially regulated in roots where highest expression was detected in response to P deficiency in the advantageous group (Fig 5). For Os11g0645800 (NB-ARC domain) patterns between groups were opposite in shoot and root and again, highest expression was detected in response to P deficiency in roots of the advantageous group. Similar strong responses to P deficiency in the advantageous allele group was seen in shoot tissue for two candidates at *qLFT-5*, WRKY (Os05g0322900) and Cytochrome P450 (Os05g0320700).

## Development of a cross population

Rice accessions harboring both advantageous alleles for panicle weight at *qLFT-5 and qLFT-11* were identified and accession GP-1103 (IRIS 313–11949) with high average total panicle weight (30.0 g plant$^{-1}$), medium heading (92 days under P deficiency) and plant height (98 cm) was selected as candidate donor. Recommended Malagasy variety X265 did not harbor either advantageous allele for panicle weight (S7 Table) and was therefore selected as the recipient parent. A set of 350 F4 lines was phenotyped on-farm under low-input conditions and wide segregation for total panicle weight per plant was observed, ranging from 13 g plant$^{-1}$ to 50 g plant$^{-1}$, which compares to 22.8 g plant$^{-1}$ for local parent X265 (Fig 6).

## Discussion

In lowland rice fields of Madagascar, the deficiency for P is typically the most serious yield-limiting factor [30], however, deficiencies for N and to a lesser extent for S and other nutrients are also common [31]. We have conducted all our field experiments on small-holder farms in fields that never received mineral fertilizer, and to which manure had not been applied at least in the two seasons preceding our experiments. Fields were therefore characterized by low fertility (Ferralsols containing very low available soil P) [7], and the average panicle weight of 16.3 g per hill, resulting in an estimated grain yield of about 3.6 t ha$^{-1}$, was just slightly above the national average of 2.9 t ha$^{-1}$ [3]. Considering that yield estimated from single row measurements tend to overestimate achievable grain yields on a field-scale, we may conclude that our field experiments represented typical low-input field conditions for the country and that genotypic differences in yield may reflect adaptations to low soil fertility. We therefore chose to designate identified QTL as *qLFT* (Low Fertility Tolerance) to distinguish the present study from field experiments conducted specifically under P deficiency (with other nutrients supplied through fertilization) and, especially to distinguish from the many QTL identified in studies conducted under controlled conditions in low-P nutrient solution.

### Loci associated with tolerance to low soil fertility

The GWAS analysis identified a highly significant locus for plant height at 38.7 Mbp on chromosome 1 (*qHt-1* in Table 1), which is only 0.3 Mbp from the position of the semidwarf gene

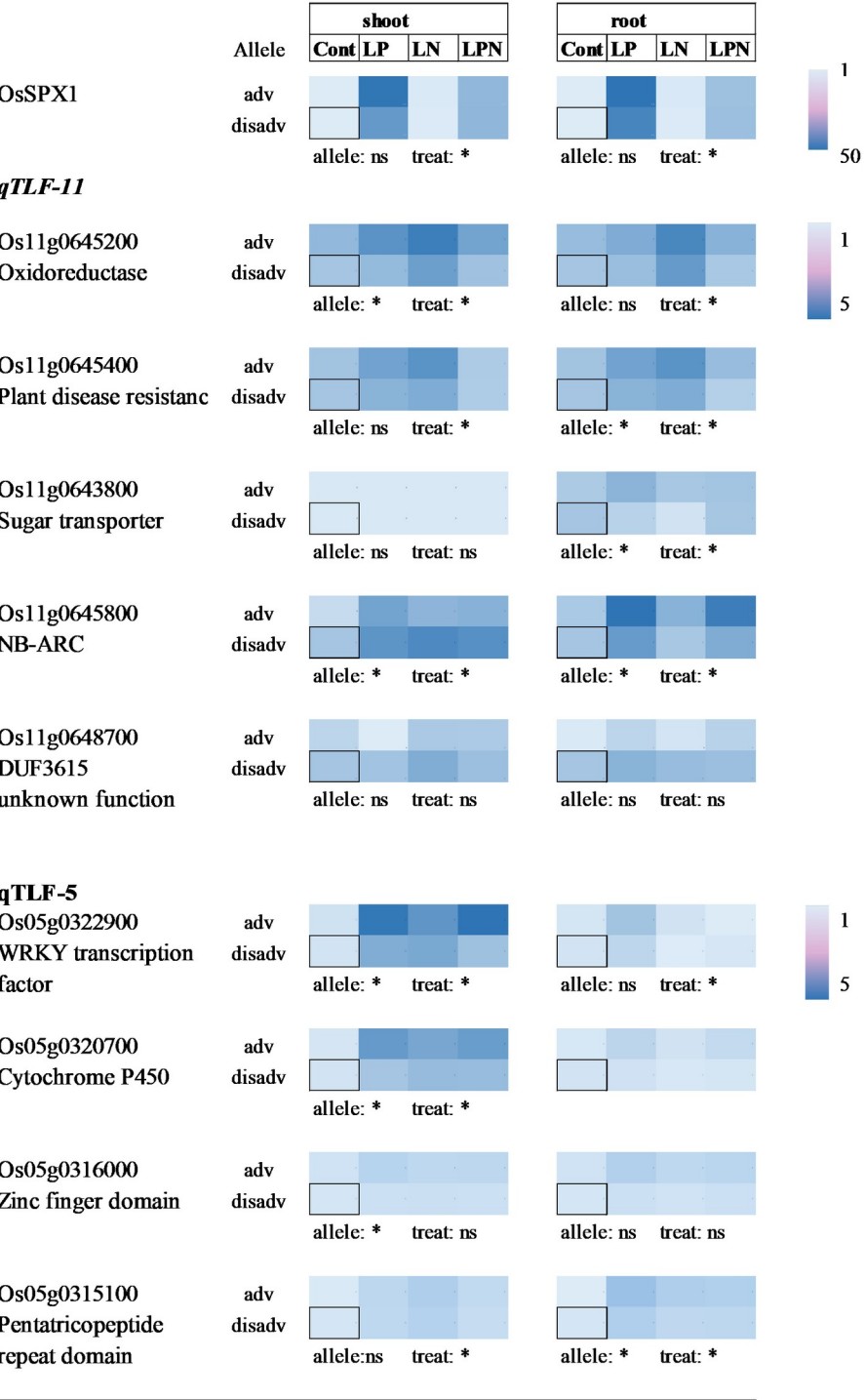

**Fig 5. Relative expression pattern of potential candidate genes located within the total panicle weight (TPW) QTLs.** Heatmap displays the differentially expressed candidate genes in root and shoot tissue from genotypes harboring advantageous/disadvantageous alleles, and under P, N, or both deficiency condition. Statistical significance was determined by two-way ANOVA and Tukey's tests. Asterisks indicates significance levels (***, **, *, ns: p<0.001, 0.01, 0.05, non-significant, respectively). Values were normalized to control treatment in accession with disadvantageous allele, in each tissue (square). adv: advantageous allele (G-38, G-355, G-1103), disadv: disadvantageous allele (X265, IR64, G-61, G-97).

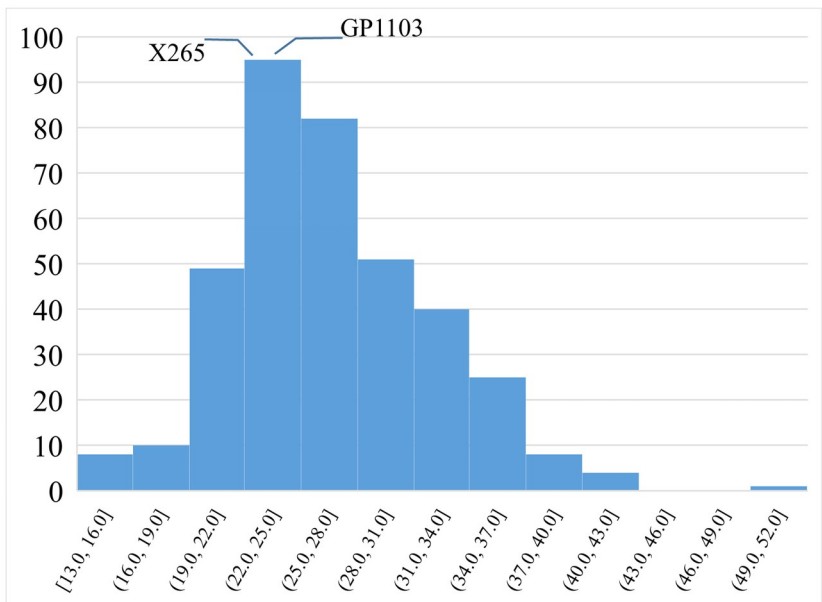

**Fig 6. Histogram showing the frequency distribution for total panicle weight among 340 lines of the X265 x GP1103, F5 population.** Plants were evaluated under low input condition in the central highland of Madagascar. The mean of GP1103 and X265 are shown in callouts.

*sd1* (Os01g0883800). As expected for a panel consisting of gene bank accession mostly exhibiting the plant habitus of traditional varieties, the minor allele (MAF = 0.11) reduced plant height by 30%, which would be consistent with most traditional varieties carrying the functional *SD1* allele. A second known locus identified in our panel through GWAS (*qHD-6* in Table 1) was within 0.3 Mbp of the known heading date gene *Hd1* (Os06g0275000) on chromosome 6. Having detected two known major genes corroborated the general suitability of the collected data for the purpose of identifying genetic determinants associated with traits of interest.

We detected two novel QTL associated with total panicle weight (TPW) and three associated with straw weight (STW) and in all cases a rare minor allele with MAF between 4–11% increased TPW and STW. These low MAF were not caused by some bias in the selection of the accessions to be phenotyped in Madagascar but could be verified among the entire 3K SNP-seek dataset (S4 Table), which showed that *qLFT-5* was even less frequent in the entire set (MAF = 3.6%) than in the phenotyped set (MAF = 5.5%). For *qLFT-11* both frequencies were around 7%. These results confirm the power of GWAS to identify rare but positive alleles in gene banks and of donors carrying such rare alleles. For the phenotyped accessions we investigated whether the origin of both rare alleles was linked to some country or region. For *qLFT*-5 all accessions belonged to the *indica* sub-species and overrepresented countries were India, Lao, and Indonesia (data not shown). For *qLFT*-11 accessions also belonged to the *indica* sub-species and overrepresented countries were China, the Philippines and Lao. Among accessions were few modern varieties such as PSBRC18, BR11 and four breeding lines from IRRI (data not shown).

## Donors and their use in rice improvement

That *qLFT-5* and *qLFT-11* headed slightly earlier than the average whereas loci associated with STW caused a delay in heading and a slight decrease in TPW indicated that the utility of STW

loci identified here for rice breeding in our target environment (highlands of Madagascar) is very limited. Late heading exposes a crop to cold spells at the end of the cropping season and can severely reduce yields. Furthermore, late heading may increase vulnerability to climate change as rainfall patterns become less predictable. Thus, we only consider the TPW loci identified here as being of interest for rice improvement.

Among the nine potential donors carrying both positive alleles, accession Liu He Xi He from China (IRIS 313–11949; ind1A) combined high TPW with medium-early heading and the medium plant height preferred in the highlands of Madagascar. A cross population between this donor and the local cultivar X265 (lacking both positive alleles) is now being evaluated under P deficiency at several sites in Madagascar in order to select breeding lines combining high grain yield with local adaptation. Since TPW was affected by environment condition ($H^2$ = 0.3), breeding lines will be tested in multi-location trials characterized by multiple nutrient deficiencies. Selected elite breeding lines could thus contribute to achieve sustainable rice production and improved food security in Madagascar.

### Putative candidate genes

The objective of this study was to evaluate a diverse panel of gene bank accessions to identify potential donors and markers to be used in rice breeding and a detailed analysis of candidate genes is beyond the scope of this study. However, patterns observed in our gene expression analysis provided some preliminary evidence suggestive of allelic differences, especially for the WRKY transcription factor and the member of the cytochrome P450 gene family on chromosome 5. For *qLFT-11* on chromosome 11 a higher proportion of differential regulation was seen in root tissue. Os11g0645800 containing the AB-ARC domain more typically associated with disease resistance [32] was strongly up-regulated by P deficiency in the advantageous allele group. Disease is unlikely to have played a role in our nutrient solution experiment, however, disease resistance would be achieved through triggered cell death [32], and one may speculate whether this could play a role in aerenchyma formation as more rapid aerenchyma formation in P efficient rice genotypes were previously reported [33].

The higher expression of sugar transporter Os11g0643800 under P deficiency in root but not shoot tissue of the positive allele group may corroborate results from the nutrient solution experiment that showed an increase root to shoot ratio of this group under P deficiency. To what extent this may be related to a shift in resource allocation to roots is a potential topic for further investigation.

### Conclusions

Alleles absent from the modern rice breeding gene pool but present in traditional varieties housed in crop gene banks have the potential to improve crop yields in less favorable environments, thereby closing the yield gap that is so persistent in Africa. However, gene banks remain a largely untapped resource and here we have attempted to address this issue by testing 500 gene bank resources for which sequence information is available. With phenotyping done in the target environment, smallholder farms under typically practice low-input conditions, our results provide evidence of power of such a GWAS approach to identify rare positive alleles in gene banks. With agronomically acceptable donors for such rare alleles identified, they may (re)-enter the breeding gene pool and contribute to variety development that would specifically benefit resource-poor farmers that have not sufficiently profited from current mainstream rice breeding.

## Supporting information

**S1 Fig. Photo showing land preparation, indirect sowing, plant growth and harvest of on-farm experiments in the central highland of Madagascar.**
(TIF)

**S2 Fig. Manhattan plots derived from GWAS analysis for traits: A) Root length, B) Root dry matter, and C) total dry matter in ratio values of treatments: Low-S/high-S.** Manhattan plot shows negative logarithmic ((-log10 (P)) values of association for each SNP (Y axis), and SNP location along the 12 chromosomes (colored bar in X axis). Red line indicates a -log10 (P value) threshold of 5.
(TIF)

**S3 Fig. Linkage Disequilibrium (LD) block for total panicle weight (TPW).** Blue lines indicate the delineated region.
(TIF)

**S4 Fig. Heatmap showing differential gene expression during vegetative and development stages reported in RiceXPro.** Veg: vegetative; repr: reproductive; LP: low Phosphorus; LN: Low Nitrogen condition.
(TIF)

**S5 Fig. Effect of variants on genes, transcripts, and protein sequence, as well as regulatory regions determined by Variant Effect Predictor (VEP, ensemble).**
(TIF)

**S1 Table. List of primers used in this study.**
(DOCX)

**S2 Table. Descriptive statistics and summary of phenotypic traits in on-farm trials (Experiment1).**
(DOCX)

**S3 Table. Complete list of QTLs associated with tolerance to low fertility soil.**
(DOCX)

**S4 Table. Allele frequency of total panicle weight (PWT) and straw dry weight (SDW) in the 3K-Rice Genome Project (3KRGP).**
(DOCX)

**S5 Table. Complete list of gene models included in GWAS associated loci.**
(DOCX)

**S6 Table. Descriptive statistics and summary of validation trial—Phenotypic traits (Experiment 2–2).**
(DOCX)

**S7 Table. Allelic distribution between the donor and recipient for the total panicle weight (PWT) QTL.**
(DOCX)

## Acknowledgments

The author would like to thank the IRRI gene bank for providing seeds of the accession used in this study.

## Author Contributions

**Conceptualization:** Juan Pariasca-Tanaka, Ryokei Tanaka, Matthias Wissuwa.

**Data curation:** Mbolatantely Fahazavana Rakotondramanana, Sarah Tojo Mangaharisoa, Harisoa Nicole Ranaivo.

**Investigation:** Juan Pariasca-Tanaka, Mbolatantely Fahazavana Rakotondramanana, Sarah Tojo Mangaharisoa, Harisoa Nicole Ranaivo, Matthias Wissuwa.

**Writing – original draft:** Juan Pariasca-Tanaka.

**Writing – review & editing:** Juan Pariasca-Tanaka, Ryokei Tanaka, Matthias Wissuwa.

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
