## [Decision Letter · Decision Letter 0]

12 Nov 2021

PONE-D-21-26210Phenotyping of a rice (Oryza sativa) association panel on smallholder farms under low-input conditions in Madagascar identifies loci associated with tolerance to low soil fertilityPLOS ONE

Dear Dr. Wissuwa,

Thank you for submitting your manuscript to PLOS ONE. After careful consideration, we feel that it has merit but does not fully meet PLOS ONE’s publication criteria as it currently stands. Therefore, we invite you to submit a revised version of the manuscript that addresses the points raised during the review process.

We look forward to receiving your revised manuscript.

Kind regards,

Muhammad Abdul Rehman Rashid, PhD

Academic Editor

PLOS ONE

Journal Requirements:

2. Please modify the title to ensure that it is meeting PLOS’ guidelines (https://journals.plos.org/plosone/s/submission-guidelines#loc-title). In particular, the title should be "specific, descriptive, concise, and comprehensible to readers outside the field" and in this case it is not informative and specific about your study's scope and methodology

Additional Editor Comments (if provided):

The presented research is interesting, but there are a lot of corrections may required before acceptance.

You may avoid the headings like "Experiment 1" etc., and indicate the theme of experiment in heading.

Please improve the drafting as per author's guide lines.

Reviewers' comments:

Reviewer's Responses to Questions

**Comments to the Author**

1. Is the manuscript technically sound, and do the data support the conclusions?

Reviewer #1: Yes

Reviewer #2: Yes

2. Has the statistical analysis been performed appropriately and rigorously? 

Reviewer #1: Yes

Reviewer #2: Yes

3. Have the authors made all data underlying the findings in their manuscript fully available?

Reviewer #1: Yes

Reviewer #2: Yes

4. Is the manuscript presented in an intelligible fashion and written in standard English?

Reviewer #1: Yes

Reviewer #2: No

5. Review Comments to the Author

Reviewer #1: The manuscript entitled: Phenotyping of a rice (Oryza sativa) association panel on smallholder farms under low-input conditions in Madagascar identifies loci associated with tolerance to low soil fertility has high significance for the researchers working on rice. Title of the manuscript would be better if the authors can avoid the repetitive word such as low. The authors have defined the objectives of the study very well. Overall the manuscript is well structured, experiments and analyses have performed very well. In abstract section, I suggest authors to include the names of the traits studied and also the names of donor and recipient parent to increase the impact of the manuscript. In this study, 532 rice accessions were study from 3K rice genome project but authors did not mention any selection criteria of these accessions. Therefore, one or two sentences explaining the selection criteria should be added.

Materials and methods section needs further improvement by adding the names of all the studied traits at one place followed by their individual explanation. Authors claim the area under study as low fertility, but without determining the soil properties of the land how it can be referred as low fertility therefore, authors are suggested to incorporate soil nutrients data particularly for N and P.

Please read line no 144 – 145 “A matrix genotype file composed of 186,229 (187K) SNPs and 3026 accessions was prepared and reported in a previous study”

Here the authors are claiming to use an already prepared matrix of 3026 accessions whereas in this study only 532 accession were included therefor it would be better to explain the reasons.

Overall manuscript reads very well. Results are well explained and validated. The results are original to merit publication. Candidate genes/QTLs identified in this study have been well characterized and being used in MAS. Therefore, after incorporating the highlighted minor correction I recommend this manuscript for publication.

Reviewer #2: Manuscript has potential for publication and having the scope for scientists community. In addition to phenotypic data analysis, heritability analysis estimates will be more fruitful for the elaborating the studied results. Find the attached file for minor sevisions

6. PLOS authors have the option to publish the peer review history of their article (what does this mean?). If published, this will include your full peer review and any attached files.

Reviewer #1: **Yes: **Dr. Muhammad Qadir Ahmad

Reviewer #2: No

---

## [Author Response · Author response to Decision Letter 0]

27 Dec 2021

RESPONSE TO EDITOR/REVIEWERS

1. Questions from Reviewer 1

Q1. Title of the manuscript would be better if the authors can avoid the repetitive word such as low. 

AU: Title was shortened as suggested. 

Q2. In abstract section, I suggest authors to include the names of the traits studied and also the names of donor and recipient parent to increase the impact of the manuscript. 

AU: names of traits and donor/recipient are included as suggested (now L25 and L26, respectively).

Q3. In this study, 532 rice accessions were study from 3K rice genome project but authors did not mention any selection criteria of these accessions. Therefore, one or two sentences explaining the selection criteria should be added.

AU: a paragraph including additional explanation was added as suggested (now L101) 

Q4. Materials and methods section needs further improvement by adding the names of all the studied traits at one place followed by their individual explanation.

AU: Studied traits were added as suggested (now L131).

Q5. Authors claim the area under study as low fertility, but without determining the soil properties of the land how it can be referred as low fertility therefore, authors are suggested to incorporate soil nutrients data particularly for N and P.

AU: The soil chemical characteristics were added as suggested (now L118)

Q6. Please read line no 144 – 145 “A matrix genotype file composed of 186,229 (187K) SNPs and 3026 accessions was prepared and reported in a previous study” 

Here the authors are claiming to use an already prepared matrix of c whereas in this study only 532 accession were included therefor it would be better to explain the reasons.

AU: We have used a previously reported matrix genotype (3026 accessions and 186,229 SNPs) because our GWAS in-house R script was based on this matrix. A subset containing the 532 accessions was then filtered, prior to the association analysis.

A sentence: “A subset containing the 532 accessions was filtered from the matrix prior to analysis” was added (now L156)

Questions extracted from comments included in the manuscript

Q7. Complete scientific name:

AU: Corrected as suggested

Q8. Please reconsider (or). I think it should be and if both heading date and plant height are not same (L25)

AU: Amended as suggested

Q9. Was this deficiency of P and N artificially induced or natural? (L28)

AU: The experiment related to deficiency of P and N deficiency was conducted under hydroponic conditions. The treatment was set to 280 mM N and/or 5 uM P, using one strength Yoshida solution as reference.

Q10. Please write the names of donor and recipient parent (L30)

AU: Please refer to Q2.

Q11. Change increasingly to unfortunately (L42)

AU: corrected as suggested.

Q12. Change much to most (L42)

AU: amended as suggested.

Q13. Rephrase (L46)

AU: modified as suggested. Now it reads: “Rice productivity in most of the SSA region is limited by several biotic and abiotic stresses. Among the abiotic stresses, low soil fertility is the one of main concern, with phosphate (P) often being the most limiting nutrient [5].” Now L45. 

Q14. Was there any basis of this selection of accessions? (L97) Pakistan is among the best rice producing countries. Is there any genotype from Pakistan included in this study? If yes please mention

AU: please refer to Q3. 

There was one accession from Pakistan, with ID: IRIS_313_8398, and name: KHARSU 80. 

Q15. Please mention year when the experiment was conducted

AU: the year has been added as suggested (now L111)

Q16. As authors claimed that fields have no history of fertilizer application eventhough, authors need to determine the soil properties and composition i.e minerals such as phosphorous. In absence of data regarding quantity of micro and macronutrients how one can claim an area as a low fertility area? (L113)

AU: please refer to Q5

Q17. Which parameters were recorded? Please enlist all once then give detail (L126)

AU: please refer to Q4

Q18. randomized complete block design (RCBD) (L204)

AU: corrected as suggested (now L216)

Q19. CV value is higher than normal? Please explain reasons (L255)

AU: a sentence “which indicates great accession variability in their adaptation to the new environment” was added as suggested (L269)

Q20. If authors agree? Its better to give name to all QTLs detected with their corresponding trait abbreviation as qHD for heading date

AU: Although we acknowledge your concern, we would prefer to keep the name of the QTLs as they are, to distinguish our result from those found in studies conducted under controlled conditions such as fertilized soils and/or under low-P nutrient solution.

2. Questions from Reviewer 2

In addition to phenotypic data analysis, heritability analysis estimates will be more fruitful for the elaborating the studied results.

AU: Thank you very much for your advice. Heritability analysis was included as complement of our result as suggested. We therefore will take into consideration these values for the development of breeding lines under multi-environmental conditions.

Several paragraphs were added in in text mentioning the heritability values (L148) in Material and Methods, (L280) in result, (L516) in discussion.

Questions/comments extracted from pdf

Q1. Replace yields to yield (L21)

AU: corrected as suggested

Q2. Provide reference about this level (L153)

AU: reference was added as requested (now L164)

Q3. Missing value at which percentage level excluded (L160)

AU: a sentence “SNP having more than 5% missing data or minor allele frequency (MAF) below 0.03 were excluded” was added as suggested (now L171)

3. Questions from Editor

Q1. Please ensure that your manuscript meets PLOS ONE's style requirements, including those for file naming. 

AU: manuscript was modified/amended following the style requirements.

Q2. modify the title to ensure that it is meeting PLOS’ guidelines 

AU: modified as suggested

Q3. We note that the grant information you provided in the ‘Funding Information’ and ‘Financial Disclosure’ sections do not match

AU: corrected as suggested

Q4. You may avoid the headings like "Experiment 1" etc., and indicate the theme of experiment in heading.

AU: corrected as suggested

---

## [Editor Report · Decision Letter 1]

3 Jan 2022

Phenotyping of a rice (Oryza sativa L.) association panel identifies loci associated with tolerance to low soil fertility on smallholder farm conditions in Madagascar

PONE-D-21-26210R1

Dear Dr. Wissuwa,

We’re pleased to inform you that your manuscript has been judged scientifically suitable for publication and will be formally accepted for publication once it meets all outstanding technical requirements.

Kind regards,

Muhammad Abdul Rehman Rashid, PhD

Academic Editor

PLOS ONE
---

## [Editor Report · Acceptance letter]

31 Jan 2022

PONE-D-21-26210R1 

*Phenotyping of a rice (Oryza sativa L.) association panel identifies loci associated with tolerance to low soil fertility on smallholder farm conditions in Madagascar*

Dear Dr. Wissuwa:

I'm pleased to inform you that your manuscript has been deemed suitable for publication in PLOS ONE. Congratulations! Your manuscript is now with our production department. 

Kind regards, 

on behalf of

Dr. Muhammad Abdul Rehman Rashid 

Academic Editor

PLOS ONE